# An Adrenalectomy Mouse Model Reflecting Clinical Features for Chronic Fatigue Syndrome

**DOI:** 10.3390/biom10010071

**Published:** 2020-01-01

**Authors:** Jin-Seok Lee, Yoo-Jin Jeon, Samuel-Young Park, Chang-Gue Son

**Affiliations:** 1Institute of Traditional Medicine and Bioscience, Dunsan Hospital of Daejeon University, Daejeon 34323, Korea; neptune@dju.ac.kr (J.-S.L.); jyj940916@naver.com (Y.-J.J.); 2Korean Medical College of Daejeon University, 62, Daehak-ro, Dong-gu, Daejeon 34323, Korea; kiseobo96@gmail.com

**Keywords:** chronic fatigue syndrome, animal model, lipopolysaccharide, polyinosinic: polycytidylic acid, adrenalectomy

## Abstract

Chronic fatigue syndrome (CFS) is one of the most intractable diseases and is characterized by severe central fatigue that impairs even daily activity. To date, the pathophysiological mechanisms are uncertain and no therapies exist. Therefore, a proper animal model reflecting the clinical features of CFS is urgently required. We compared two CFS animal models most commonly used, by injection with lipopolysaccharide (LPS from *Escherichia coli* O111:B4) or polyinosinic: polycytidylic acid (poly I:C), along with bilateral adrenalectomy (ADX) as another possible model. Both LPS- and poly I:C-injected mice dominantly showed depressive behaviors, while ADX led to fatigue-like performances with high pain sensitivity. In brain tissues, LPS injection notably activated microglia and the 5-hydroxytryptamine (HT)_1A_ receptor in the prefrontal cortex and hippocampus. Poly I:C-injection also remarkably activated the 5-HT transporter and 5-HT_1A_ receptor with a reduction in serotonin levels in the brain. ADX particularly activated astrocytes and transforming growth factor beta (TGF-β) 1 in all brain regions. Our results revealed that LPS and poly I:C animal models approximate depressive disorder more closely than CFS. We suggest that ADX is a possible method for establishing a mouse model of CFS reflecting clinical features, especially in neuroendocrine system.

## 1. Introduction

Chronic fatigue syndrome (CFS) is an invisible disability that is mainly identified by extreme fatigue that is unexplained medically. More than 25% of CFS patients are bedridden or house-bound all day [1], and they have a far lower health-related quality of life (QoL) than individuals with cancer, diabetes, or stroke [2,3]. People suffering from CFS are estimated to be 0.8–3.5% of the general population worldwide [4]. Until now, there has been no objective diagnostic marker, and the etiology is uncertain, which leads to difficulty in developing therapies. Accordingly, many researchers and health-related national policies have emphasized studies to elucidate the molecular mechanisms of CFS [5,6].

The pathological mechanisms of CFS have been suggested from wide aspects, including immunology, metabolism, neurobiology, endocrinology, and psychology [7,8]. Single contributing cause such as virus infection is being proposed until recently [9], however the most convincing hypothesis is a complex multisystemic disorder caused by neuro-psycho-endocrine-immunological dysfunction [10,11]. Several studies proposed the serotonergic hyperactivity and hypocortisolism as features in patients suffering CFS [12,13,14]. Recently, systematic review and cross-sectional study revealed the notable increase of circulating TGF-β (a systemic immunosuppressor) levels in CFS patients [15,16]. It is well known that glucocorticoids inhibit production of active TGF-β [17]. These clinical findings should be supported by animal-based studies to facilitate the exploration of CFS pathophysiology. However, there is no available animal model properly reflecting the clinical features of CFS. To develop the representable CFS animal model is an important issue in processes of the in-depth study for pathology and researches for its therapeutics.

At present, most of the adapted animal models for CFS are derived from gram-negative endotoxin (lipopolysaccharide (LPS) and *Brucella abortus*) injection, immunostimulant (poly I:C) treatment, or chronic stress-exposure [18,19]. Both the LPS and poly I:C models are believed to mimic the immunopathological relevance of CFS [20,21] while chronic stress-subjected animal models mimic the conditions of pathologies in the neurological or endocrine systems [22]. These animal models, however, are questionable due to a lack of proper application to CFS in terms of clinical and neuro-molecular findings.

We hereby evaluated the appropriateness of LPS- and poly I:C-derived models in comparison to clinical features of CFS. To facilitate the development of suitable animal models for CFS, we propose that bilateral adrenalectomy (ADX) approach can be considered as an available CFS mouse model in terms of hypocortisolism in CFS patients.

## 2. Material and Methods

### 2.1. Materials

The following reagents were obtained from Sigma-Aldrich (St. Louis, MO, USA); glycerol, 4′,6-diamidino-2-phenylindole dihydrochloride (DAPI), aqueous mounting buffer, sodium hydroxide, tetramethylethylenediamine (TEMED), copper(II) sulfate solution, LPS from *Escherichia coli* O111:B4, sucrose, Tween 20, and bovine serum albumin (BSA). Other reagents were obtained from the following manufacturers: Triton X-100, paraformaldehyde powder, acetyl alcohol, hydrogen peroxide (H_2_O_2_), methylene alcohol, butanol, isopentene, sulfuric acid, phosphate, and ethylene glycol (Duksan Science, Seoul, Korea; Daejung Chemicals & Metals Co., Siheung, Korea; Junsei Chemical Co., Ltd., Tokyo, Japan), optimal cutting temperature (OCT) compound, protease inhibitor, normal goat serum, 5-hydroxytryptamin (5-HT), tryptophan hydroxylase 2 (TPH2), glial fibrillary acidic protein (GFAP), 5-HT transporter, 5HT_1A_ receptor, β-actin, and fluorescence- and horseradish peroxidase (HRP)-conjugated secondary antibodies (Abcam, Cambridge, MA, USA; Thermo-Fisher Scientific, Allentown, PA, USA; Santa Cruz Biotechnology, Dallas, TX, USA); and AMRESCO and RNAlater (Ambion, Austin, TX, USA).

### 2.2. Animals and Experimental Design

One hundred thirty-two specific pathogen-free male ICR mice (8 weeks old, 32–35 g) were obtained from Dae-Han Bio link Co., Ltd. (Eumseong, Korea). Mice were housed in an environmentally controlled room maintained using a thermohygrostat (ALFFIZ, BuSung Co., Ltd., Seoul, Korea) at 22 ± 2 °C and 60% ± 5% relative humidity under a 12/12 h light/dark cycle. The mice were fed a commercial standard chow diet (Daehanbio-link) with tap water ad libitum.

After acclimation for 7 days, mice were allocated into four groups (vehicle, LPS, poly I:C, and adrenalectomy (ADX)) for six sets of the following experiments: open field test (*n* = 24, six mice per group), rotarod test (*n* = 24, six mice per group), grip strength test and plantar test (*n* = 24, six mice per group), passive avoidance test (*n* = 24, six mice per group), serum biochemistry and Western blot analysis (*n* = 24, six mice per group), and immunohistological staining analysis (*n* = 12, three mice per group; Appendix A). Normal saline, LPS (3 mg/kg) or poly I:C (20 mg/kg) was intraperitoneally injected for the corresponding groups, 24 h (normal saline and LPS) or 4 h (poly I:C), before behavioral tests or sacrifice. Regarding the ADX model, bilateral adrenalectomy was performed under anesthesia with ketamine (100 mg/kg), according to a previous method [23]. Under anesthesia, the small incision was made in the skin near each kidney. The both adrenals were removed from the surrounding fat, and then adrenalectomized mice were immediately put on a heating chamber. To aid healing, the povidone-iodine was daily applied to wound regions and supplied with 0.9% saline and 1% glucose in tap water to compensate the loss of aldosterone. These mice were then allowed to recover for 7 days as previously described [24].

Animal care and experiments were conducted in accordance with the guidelines issued by the Institutional Animal Care and Use Committee of Daejeon University (Daejeon, Republic of Korea; Approval No. DJUARB 2018-018) and the Guide for the Care and Use of Laboratory Animals published by the United States National Institutes of Health.

### 2.3. Evaluation of Anxiety and Locomotor Behavior, Open Field Test

The open field test was conducted as previously described with slight modification [25]. The plastic enclosure box for the open field apparatus was contained in the black square side (40 cm × 40 cm × 30 cm), and the center of the field was distinguishable in the recording software. Before beginning the trial, mice were acclimated in the testing room for 30 min. Behavioral testing was conducted twice for 5 min (interval 1 h) at 7–8 lux illumination. The parameters, including the time spent in the center zone, zone transition number, and total distance, were recorded using a video camera connected to the corresponding software (Smart Junior, Panlab SL, Barcelona, Spain).

### 2.4. Evaluation of Motor Activity, Rotarod Test

The motor activity was evaluated using a rotarod machine (ENV-574M, Med Associates Inc., St. Albans, VT, USA) according to manufacturer’s instructions. Before the training test, the mice were habituated to stay on the stationary drum (4 rpm, 1.12 m/min on the surface) for 3 min. The mice were placed back on the drum immediately if they fell, up to five times during habituation. After habituation for 3 days, the motor activity was evaluated (three trials, interval 20 min) on the drum accelerated from 4 to 40 rpm. The latency to fall was recorded. 

### 2.5. Evaluation of Muscular Strength, Grip Strength Test

The muscular strength was evaluated using a grip strength test apparatus (BIO-G53, BIOSEB, Pinellas Park, FL, USA) connected with a wire grid (9 cm × 15 cm) and an isometric force transducer (dynamometer). The mice were allowed to grasp the grid with their forepaws, and gently pulled backward until losing grip within 3 s. The grip strength was evaluated (three trials, interval 5 min), and then the mean of maximal force was expressed in Newtons (N).

### 2.6. Evaluation of Pain Sensitivity, Plantar Test

The pain sensitivity was tested using a Hargreaves apparatus (37370-002, Ugo Basile, Comerio, Italy) according to manufacturer’s instructions. The mice were placed in individual plexiglass cubicles (8.5 cm × 3.4 cm × 3.4 cm) and allowed to acclimatize for 15 min before the trial. A constant-intensity radiant heat source (beam diameter 0.5 cm and intensity 70 I.R.) was aimed at the midplantar area of the hind paw. The latency time (seconds) until paw withdrawal was recorded.

### 2.7. Evaluation of Long-Term Memory, Passive Avoidance Test

Fear-conditioning learning and memory were evaluated using a step-through passive avoidance apparatus (JD-SI-10, JEUNGDO Bio & Plant Co., Ltd., Seoul, Korea) with a two-compartment box (one light and one dark compartment connected by a sliding door). In the acquisition trial, the mice were punished by electrical shock (0.4 mA) for 5 s when they entered the dark compartment. After 24 h, each mouse was replaced in the light compartment for retention trials. The latency time to enter the dark compartment was recorded for 5 min.

### 2.8. Sample Preparation

The mice were sacrificed under CO_2_ anesthesia. The serum was collected by centrifugation at 3000 × *g* for 15 min. The brain was immediately removed and specific regions were isolated using a coronal mouse brain matrix (1 mm, BSMAS001-1; Zivic Instruments, Pittsburgh, PA, USA) and biopsy punch (1 mm, BP-10F, Kai Medical, Seki, Japan). The tissues of each region were homogenized in a radioimmunoprecipitation assay (RIPA) buffer supplemented with protease inhibitors. For the immunohistological analysis, the mice were subjected to transcardial perfusion with heparin (10 units/mL) and 4% paraformaldehyde (PFA) solution, and their brains were fixed with 4% PFA. The total protein concentrations were measured using a bicinchoninic acid protein assay kit (Sigma). The absorbance at 560 nm was measured using a UV spectrophotometer (Molecular Devices Corp., Sunnyvale, CA, USA).

### 2.9. Determination of Corticosterone and TNF-α

The serum levels of corticosterone and TNF-α were measured according to manufacturer’s instructions (K014-H5, Arbor Assays, Michigan, USA for corticosterone and 558534 BD Biosciences, San Diego, CA, USA for TNF-α). Absorbance at 450 nm was measured using a UV spectrophotometer (Molecular Devices).

### 2.10. Determination of TGF-β1

The active form of TGF-β1 in serum or brain regions, including the prefrontal cortex, hippocampus, hypothalamus, and raphe nuclei was detected according to manufacturer’s instructions (DY1679-05, R&D systems, Camarillo, CA, USA). The absorbance at 450 nm was measured using a UV spectrophotometer (Molecular Devices).

### 2.11. Western Blotting Analysis

To determine the protein levels of 5-HT transporter and 5HT_1A_ receptor in seven brain regions, brain tissues were denatured by boiling for 10 min. Then samples were separated by 10% polyacrylamide gel electrophoresis and transferred to polyvinylidene fluoride (PVDF) membranes. After blocking in 3% BSA for 1 h, the membranes were probed with primary antibodies such as 5-HT transporter (1:500, ab130130, Abcam), 5HT_1A_ receptor (1:1000, ab85615, Abcam), or β-actin (1:1000, MA5-11869, Thermo-Fisher Scientific, Allentown, PA, USA) overnight at 4 °C. The membranes were washed three times and incubated with HRP-conjugated anti-rabbit (1:2500, against 5HT_1A_ receptor), anti-goat (1:2500, against 5HT transporter), or anti-mouse antibody (1:2500, against β-actin) for 40 min. The protein was visualized using an enhanced chemiluminescence (ECL) advanced kit (Thermo Fisher Scientific, CA, USA). The protein expression was observed using the FUSION Solo System (Vilber Lourmat, Collegien, France), and its intensity was analyzed with ImageJ version 1.46 (NIH, Bethesda, MD, USA).

### 2.12. Immunohistological Staining Analysis

The brain tissues were immersed in the 4% PFA solution for 48 h, and subsequently cryoprotected in 10–30% sucrose solution for 24 h each. The brain tissues were embedded in optimum cutting temperature (OCT) compound (Leica Microsystems, Bensheim, Germany) with liquid nitrogen, and cut into coronal frozen sections (35 μm) using a cryostat (CM3050S, Leica Microsystems, Nussloch, Germany). Frozen sections of brain tissue were stored in the anti-freeze buffer. To block endogenous peroxidase activity, the free-floating sections were immersed in 1% H_2_O_2_. The sections were treated with blocking buffer (5% normal chicken serum and 0.3% Triton X-100 in cold-PBS) and incubated with Iba-1 (1:200, #019-19741, Wako, Osaka, Japan) or 5-HT (1:200, ab66047, Abcam, Cambridge, MA, USA), TPH2 (1:200, NB100-74555, Novus, St. Louis, MO, USA), and GFAP (1:200, Z0334, Dako, Carpinteria, CA, USA) primary antibodies overnight at 4 °C. After washing with ice-cold PBS, sections were incubated with a goat anti-rabbit IgG HRP (1:400, for Iba-1, GeneTex, GTX213110-01), donkey anti-goat IgG H&L (1:400, for 5-HT, Alexa Fluor 488, ab150129), goat anti-rabbit IgG H&L (1:400, for TPH2, Alexa Fluor 594, ab150080), or goat anti-rabbit IgG H&L (1:400, for GFAP, Alexa Fluor 488, ab150077) secondary antibody for 2 h at 4 °C. For the Iba-1-positive signal, the sections were exposed to an avidin-biotin peroxidase complex (PK-6200, VECTASTAIN Elite ABC kit, Vector Laboratories, Burlingame, CA, USA) for 1 h. Then, the peroxidase activity was color-developed with stable, 3′-diaminobenzidine, and immunoreactions were observed using a binocular microscope (DMLS, Leica, Germany). For the 5-HT, TPH2, and GFAP-positive signals, the sections were subsequently exposed to DAPI (1:1000, D9542, Sigma-Aldrich, St. Louis, MO, USA). The immunoreactions were observed under a fluorescence microscope (1 × 71, OLYMPUS, Tokyo, Japan), and the intensities were quantified using ImageJ 1.46 software (NIH, Bethesda, MD, USA).

### 2.13. Statistical Analysis

All data are expressed as the mean ± standard deviation (SD). The statistical significance was analyzed by one-way analysis of variance (ANOVA), followed by post hoc multiple comparisons with Tukey HSD using IBM SPSS statistics software, version 25.0 (SPSS Inc., Chicago, IL, USA). Differences at *p* < 0.05 indicate statistical significance.

## 3. Results

### 3.1. Changes on Depressive/Anxious-Like Behaviors: Open Field Test

The poly I:C injection tended to reduce locomotor activity (*F*(3,20) = 2.291, *p* = 0.109) compared with vehicle injection (*p* = 0.08, Figure 1A). The time spent in the center field (*F*(3,20) = 22.253, *p* < 0.01) was significantly shortened by LPS injection, whereas the mice subjected to ADX spent more time in the center field the vehicle-treated mice (*p* < 0.05 for LPS injection or *p* < 0.01 for ADX, Figure 1B). Furthermore, LPS injection induced a significant reduction in the zone transition numbers (*F*(3,20) = 5.907, *p* < 0.01) compared with vehicle injection, and poly I:C injection showed a decreased tendency (*p* < 0.01 for LPS, *p* = 0.09 for poly I:C, Figure 1C).

### 3.2. Changes on Fatigue-Like Behaviors: Rotarod Test

No experimental group showed significant alterations in rotarod test performance compared with the vehicle-treated group. Mice that underwent ADX surgery showed a tendency of reduced latency (*F*(3,20) = 1.041, *p* = 0.396) to fall from the accelerating rotarod compared with mice-treated with vehicle (*p* = 0.09, Figure 1D). There were no significant differences in other groups.

### 3.3. Changes on Muscular Strength and Pain Sensitivity: Grip Strength Test and Plantar Test

ADX (26%) significantly reduced the maximum grip strength (*F*(3,20) = 5.116, *p* < 0.05, Figure 1E). Moreover, the paw withdrawal latency to noxious heat (*F*(3,20) = 3.175, *p* < 0.05) was significantly decreased by ADX compared with that by vehicle treatment (*p* < 0.05 for ADX, Figure 1F). Other groups did not show significant differences in either behavioral test.

### 3.4. Changes on Long-Term Memory: Passive Avoidance Test

LPS and poly I:C injection tended to increase the latency to enter into the dark component (*F*(3,20) = 2.575, *p* = 0.083) compared with vehicle injection, but not significantly (Figure 1G).

### 3.5. Changes of the Astrocyte Activity in Brain Tissues

LPS, poly I:C and ADX markedly increased reactive astrocytes in the hippocampal subregions (DG, dentate gyrus (*F*(3,8) = 65.262, *p* < 0.01); CA3, cornu ammonis 3 (*F*(3,8) = 19.788, *p* < 0.01)) and hypothalamus (*F*(3,8) = 37.948, *p* < 0.01) compared with the vehicle. In particular, the density of reactive astrocytes was considerably higher in ADX-subjected mice than in the hippocampal subregions (two-fold for DG and 1.5-fold for CA3) of vehicle-treated mice (*p* < 0.05 for ADX). Moreover, in the case of the hypothalamus, both poly I:C and ADX significantly upregulated GFAP-positive signal compared with the vehicle (*p* < 0.05 for both, Figure 2A,B).

### 3.6. Changes of TGF-β1 in Serum and Brain Tissues

The production of the TGF-β1 active form was significantly reduced (approximately 35%) in the serum (*F*(3,20) = 17.761, *p* < 0.01) of LPS-injected mice compared with that in the serum of vehicle-injected mice (*p* < 0.05), whereas ADX slightly tended to increase the serum TGF-β1 (Figure 2C). Among four brain regions, the level of TGF-β1 was significantly increased by poly I:C (*p* < 0.05 in the raphe nuclei) and ADX (*p* < 0.05 in the hippocampus (*F*(3,20) = 11.670, *p* < 0.01), hypothalamus (*F*(3,20) = 18.444, *p* < 0.01), and raphe nuclei (*F*(3,20) = 8.497, *p* < 0.01) and *p* < 0.01 in the PFC (*F*(3,20) = 18.876, *p* < 0.01)) compared with the vehicle. However, LPS did not affect this change (Figure 2D).

### 3.7. Changes of the Microglia Activity in Brain Tissues

Among the groups, LPS remarkably activated the microglial cells in several brain regions, including the PFC (*F*(3,8) = 25.545, *p* < 0.01), CA3 of the hippocampus (*F*(3,8) = 89.390, *p* < 0.01), and hypothalamus (*F*(3,8) = 649.081, *p* < 0.01; *p* < 0.05 or *p* < 0.01), compared with the vehicle. In addition, the microglial activation was not activated by poly I:C or ADX (Figure 3A,B).

### 3.8. Changes of Corticosterone and TNF-α in Serum

The LPS and poly I:C significantly increased the serum corticosterone (two-fold), whereas ADX completely depleted the concentration of corticosterone (*F*(3,20) = 28.519, *p* < 0.01) in serum compared with the vehicle (*p* < 0.01 for all groups, Figure 3C). 

The serum level of TNF-α (*F*(3,20) = 121.300, *p* < 0.01) was significantly increased by LPS (three-fold) and poly I:C (four-fold) injection. Furthermore, ADX (1.6-fold) also significantly increased circulating TNF-α in serum compared with the vehicle (*p* < 0.01 for all groups, Figure 3D).

### 3.9. Changes of Serotonergic System in Brain Tissues

From the 5-HT (*F*(3,8) = 16.410, *p* < 0.01) and TPH2 (*F*(3,8) = 98.725, *p* < 0.01) analyses, both signals were markedly reduced in the poly I:C injection group, whereas the levels of both markers in ADX-subjected mice and TPH2 in LPS-injected mice were significantly increased in the dorsal raphe nuclei compared with the levels in vehicle-treated mice (*p* < 0.05 or *p* < 0.01, Figure 4A,B).

The protein expression of the 5-HT transporter (*F*(3,8) = 171.100, *p* < 0.01 in PFC; *F*(3,8) = 33.451, *p* < 0.01 in hippocampus; *F*(3,8) = 29.948, *p* < 0.01 in hypothalamus; *F*(3,8) = 79.746, *p* < 0.01 in raphe nuclei) was significantly increased by LPS (hippocampus and hypothalamus), poly I:C (PFC, hippocampus, and raphe nuclei), and ADX (PFC and raphe nuclei) compared with vehicle (*p* < 0.05 or *p* < 0.01, Figure 5A,B). The alterations in 5-HT_1A_ receptor expression (*F*(3,8) = 286.896, *p* < 0.01 in PFC; *F*(3,8) = 512.493, *p* < 0.01 in hippocampus; *F*(3,8) = 453.658, *p* < 0.01 in hypothalamus; *F*(3,8) = 228.000, *p* < 0.01 in raphe nuclei) were shown in mice that underwent LPS injection (PFC, hippocampus, and raphe nuclei), poly I:C injection (PFC, hippocampus, hypothalamus, and raphe nuclei), and ADX (hippocampus, hypothalamus, and raphe nuclei) compared with vehicle injection (*p* < 0.05 or *p* < 0.01, Figure 5A,C). In the hypothalamus, 5-HT_1A_ receptor expression was significantly increased 5.3-fold in ADX-subjected mice (*p* < 0.01).

## 4. Discussion

The elucidation of the molecular mechanisms of CFS is urgently needed for the discovery of therapies, and many governments in the US and Europe have encouraged research for CFS by increasing funding [26,27]. To contribute to these efforts, we evaluated the features of the LPS, poly I:C, and ADX mouse models in comparison with the major pathophysiological features of CFS.

To date, preclinical CFS studies mainly used LPS-, poly I:C-, or chronic stress-induced animal models [18,28]. For LPS model, we used LPS originated from *Escherichia coli* (O111:B4), which is known to induce the illness-like symptoms [29,30]. The LPS dose (3 mg/kg) and time duration (24 h after injection) were chosen based on the concerns about non-response versus septic shock [31,32] and peak time of locomotor impairment [33]. We excluded the stress-induced model because its aspects of endocrinology were not similar to those of CFS, just to the chronic fatigue or depression [34,35]. Although fatigue is a major symptom in both CFS and chronic fatigue, they are believed to be obviously different [36,37]. To avoid confusion in those names owing to “fatigue”, the Institute of Medicine (IOM) in the US recommended revising CFS to systemic exertion intolerance disease (SEID) in 2015 [38]. The ADX model, instead of stress-exposure, was included in the present study and reflects hypocortisolism. Hypocortisolism is well defined as a typical feature of CFS patients [39].

From our results, neither LPS nor poly I:C injection affected motor activity, muscular strength or pain sensitivity, while these features were notably reduced by ADX (even *p* = 0.09 in motor activity; Figure 1D–F). No significant difference in complete blood counts (Appendix A), body weight (Appendix A), or food intake (data not shown) was observed between sham-operated and ADX groups. Additionally, the results for depressive and anxiety-related behaviors were reversed in the LPS and poly I:C models in comparison with the ADX model (Figure 1A–C). Central fatigue is a term suitable for CFS symptoms, indicating both physical and mental disabilities such as severe fatigue, muscle pain, cognitive impairment, and unrefreshing sleep [40]. OM proposed post-exertional malaise (PEM), unrefreshing sleep, and cognitive impairment but not depression as key CFS diagnostic criteria in addition to medically unexplained profound fatigue [38]. None of the three models induced memory loss in the passive avoidance test (Figure 1G). Regarding depressive behavior, it is very prevalent (42.2%) in patients with CFS, and anti-depressant drugs are frequently prescribed to CFS patients [41]. However, these depression-related therapies were not accepted by scientists and patients [42,43]. Accordingly, the depressive phenotype of CFS would be different (hyperserotonergic form) with major depressive disorder showing serotonin depletion [44].

Serotonergic hyperactivity is a distinguishable biological feature of CFS, but not of fibromyalgia and depression [44,45]. One clinical study found altered 5-HT transporter function in the anterior cingulate cortex of CFS patients, which is inversely correlated with the pain score [12]. Furthermore, sensitivity to the 5-HT_1A_ receptor (inhibitory receptor) was noteworthily lowered in most brain regions of CFS patients [13]. These hyperserotonergic findings of CFS are in stark contrast to those of major depressive disorder [46]. As expected, the ADX model showed an increased quantity of 5-HT and its synthetic enzyme (TPH2) activity in the dorsal raphe nuclei (Figure 4A,B). However, these results were not supported by the protein assay data from the 5-HT transporter and 5-HT_1A_ receptor in brain tissues (Figure 5A–C), which seems to be a transient condition following rapid increase of serotonin. Alterations in serotonergic circulation are also associated with irritable bowel syndrome-diarrhea [47] and unrefreshing sleep [48], as common comorbidities of CFS. In contrast, poly I:C-treated mice showed typical depressive biological features, such as the hypoactivity of 5-HT and TPH2 as well as hyperactivity of the 5-HT transporter and 5-HT_1A_ receptor. The LPS injection model showed a mixed form of the ADX and poly I:C model in 5-HT-related immunohistochemistry and protein assays (Figure 4 and Figure 5). Animal and clinical studies have found that serotonergic upregulation aggravates fatigue-like symptoms [49,50,51]. One research group also found that central fatigue could be induced in serotonin spillover conditions via motoneuron inactivity [52]. Although our data could not mimic the serotonergic feature of CFS completely, the ADX model could be supported by the linkage between the fatigue-like behaviors and up-regulation of serotonin.

As expected, ADX completely depleted the serum corticosterone level, but it was considerably elevated in both the LPS and poly I:C models (Figure 2C). Hypocortisolism is an endocrinological hallmark of CFS, which is diametrically opposed to fibromyalgia and depressive disorder [53]. In patients with CFS, the low cortisol awakening response is believed to affect chronic pain [54] and cause impairment in sleep-wake cycles [55] and PEM symptoms [56]. Intriguingly, the ADX group showed increased 5-HT_1A_ receptor expression (5.3-fold) in the hypothalamus (Figure 4). This condition can suppress the hypothalamus-derived signaling downstream of the endocrine axis, causing irreversible fatigue and the lowest QoL despite resting. A previous study found that desensitization of 5-HT_1A_ receptor in hypothalamus led to decrease of adrenocorticotropic hormone response (ACTH), corticosterone-stimulating hormone [57]. This inverse interaction between glucocorticoids and serotonin may participate in the neuro-endocrinal features of CFS. ADX or treatment with a glucocorticoid receptor antagonist (RU486) inhibited exercise tolerance against weight-loaded forced swimming in mice [58].

It is known that glucocorticoid hormone is a key regulator of the inflammatory response, and previous knowledge revealed that glucocorticoids block the production of TGF-β [59]. As expected, ADX significantly facilitated the production of the TGF-β1 active form in the four brain regions and serum (increasing trend; Figure 2). The astrocyte, the main source of TGF-β in the brain, was also conspicuously activated in the hippocampus and hypothalamus of the ADX group (Figure 2). Animal studies have suggested that elevated TGF-β levels in cerebrospinal fluid are the cause of fatigue [59,60]. Recently one cross-sectional study showed that only serum TGF-β among 51 cytokine profiles had a complete elevation in CFS subjects [16], in accordance with meta-analysis data [17]. In contrast to TGF-β, a typical inflammatory cytokine, TNF-α did not show much difference in those studies. Neuroinflammation-associated microglial activation is well studied in many neuropsychiatric disorders, especially in major depressive disorder [61]. In our study, elevated serum TNF-α and activated microglia were found in both the LPS and poly I:C groups, but not in the ADX group (Figure 3). Even though brain neuroinflammation have been implied as a major molecular biomarker of CFS, it is still being debated [62].

## 5. Conclusions

Taken together, our results indicate the further feasibility of the ADX model for CFS animal studies, while both LPS and poly I:C models were indicated for neuropsychiatric disorders (Figure 6). Our study had limitations such as a lack of clinical-based pathophysiological features for reference and a comparison of the ADX group with the vehicle group, not a sham operation group. This was conducted after the confirmation of no difference in hematological analysis between vehicle and sham (Appendix A). Despite some limitations, our study is the first to comprehensively demonstrate the merits and/or demerits of the LPS, poly I:C, and ADX model with respect to the clinical features of CFS. Our data could be a reference model for the animal-based molecular mechanisms of CFS or for pharmacological studies of CFS in the future.

## Figures and Tables

**Figure 1 biomolecules-10-00071-f001:**
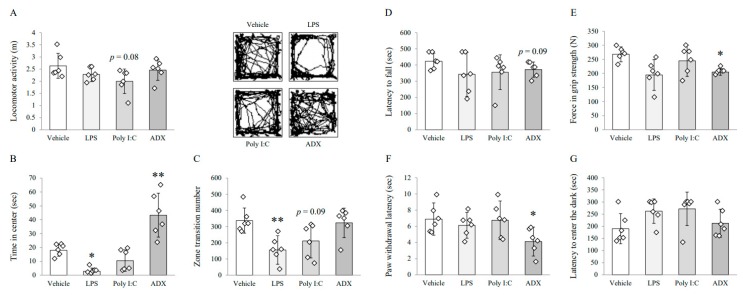
Behavioral tests for symptoms of chronic fatigue syndrome (CFS) among mice models. Locomotor activity (**A**), time spent in the center (**B**), and zone transition numbers (**C**) on the open field test for depressive-like behaviors. Latency to fall (**D**) on the rotarod test for motor activity. Force (**E**) on the grip strength test for muscular strength. Paw withdrawal latency (**F**) on the plantar test for pain sensitivity. Latency to enter the dark compartment (**G**) on the passive avoidance test for long-term memory. Data are expressed as the means ± SD (*n* = 6). *, *p* < 0.05; **, *p* < 0.01 compared with the vehicle group. ADX; adrenalectomy, LPS; lipopolysaccharide.

**Figure 2 biomolecules-10-00071-f002:**
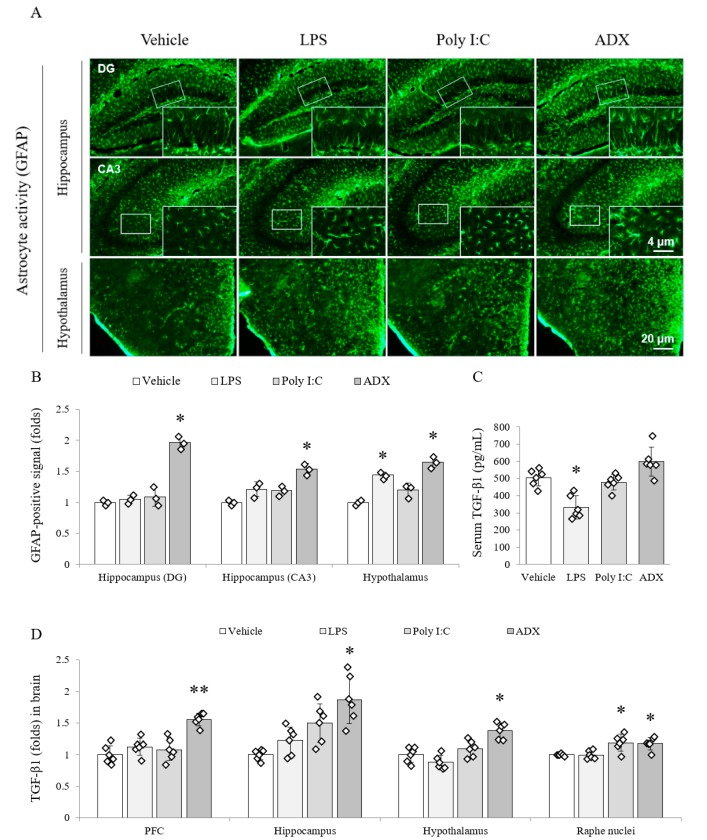
Astrocyte activity and TGF-β1 expression in both brain regions and serum. GFAP-positive stained astrocytes in hippocampal subregions and the hypothalamus (**A**). Representative photomicrographs were taken at magnifications of 100× and 500×. Quantification of astrocyte activity in brain regions (**B**). Active form of TGF-β1 in serum (**C**) and brain regions (**D**). Data are expressed as the means ± SD (*n* = 6). *, *p* < 0.05; **, *p* < 0.01 compared with the vehicle group. DG; dentate gyrus, CA3; cornu ammonis3, TGF-β1; transforming growth factor-beta1.

**Figure 3 biomolecules-10-00071-f003:**
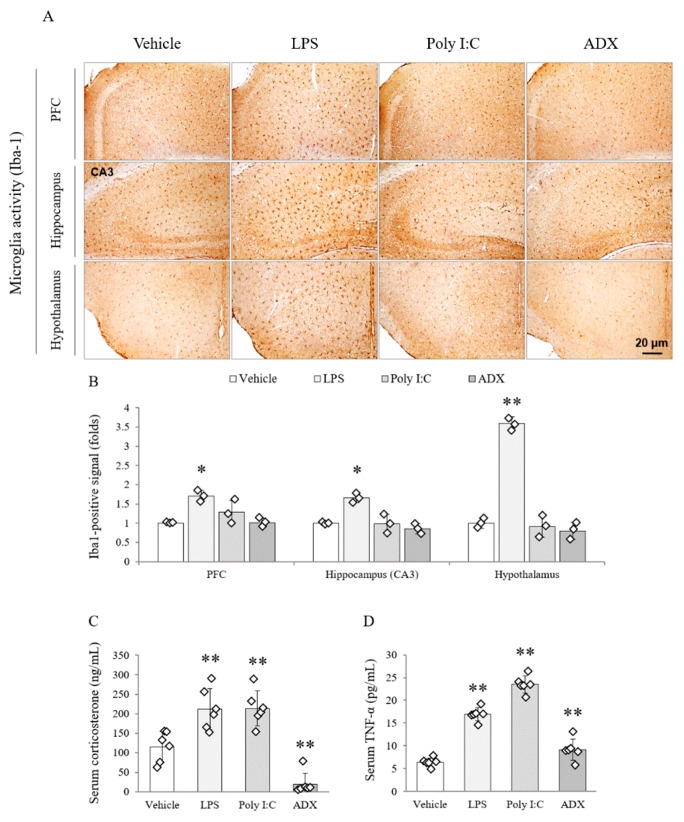
Microglial activity in brain regions and stress-related hormones and cytokines in serum. Iba1-positive stained microglia cells in the PFC, hippocampal CA3 and hypothalamus (**A**). Representative photomicrographs were taken at magnifications of 100×. Quantification of microglial activity in brain regions (**B**). Corticosterone (**C**) and TNF-α (**D**) in serum. Data are expressed as the means ± SD (*n* = 3 or 6). *, *p* < 0.05; **, *p* < 0.01 compared with the vehicle group. PFC; prefrontal cortex.

**Figure 4 biomolecules-10-00071-f004:**
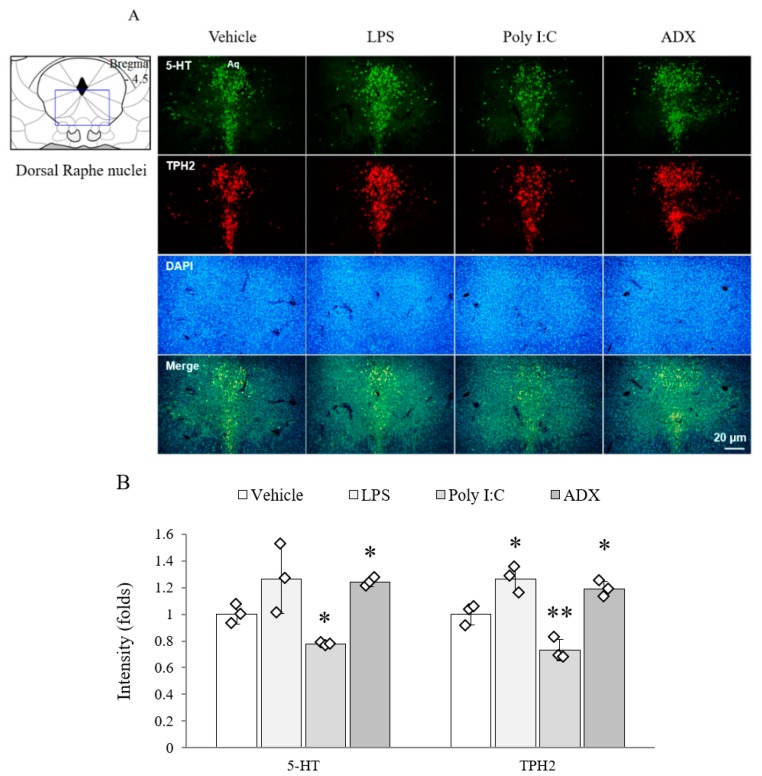
The 5-HT and TPH2 in the raphe nuclei of the brain. The 5-HT- and TPH2-positive double stained cells in the dorsal raphe nuclei (**A**). Representative photomicrographs were taken at magnifications of 100×. Quantification of 5-HT and TPH2 (**B**). Data are expressed as the means ± SD (*n* = 3). *, *p* < 0.05; **, *p* < 0.01 compared with the vehicle group. 5-HT; 5-hydroxytryptamine, TPH2; tryptophan hydroxylase, DAPI; 4′,6-diamidine-2-phenylindole dihydrochloride.

**Figure 5 biomolecules-10-00071-f005:**
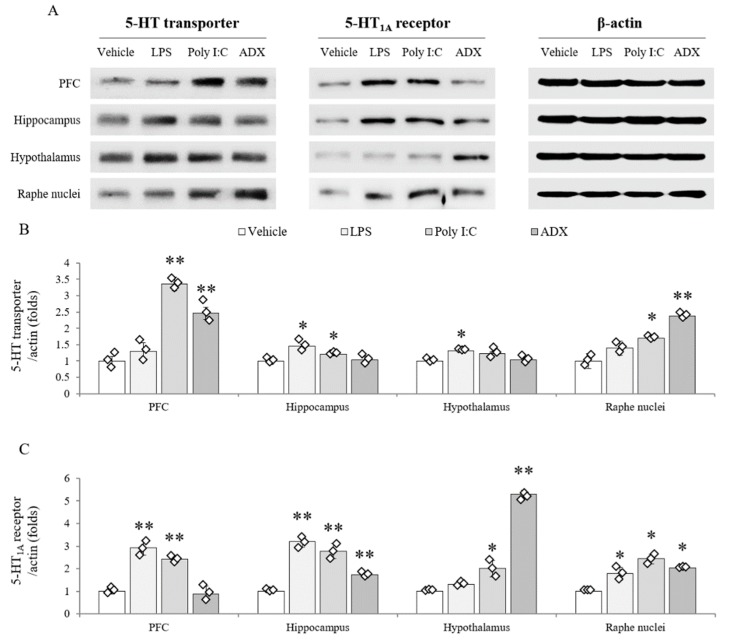
Serotonergic transporter and receptor expression in brain regions. The 5-HT transporter and 5-HT_1A_ receptor protein expression in brain regions (**A**). Quantification of 5-HT transporter (**B**) and 5-HT_1A_ receptor expression (**C**). Data are expressed as the means ± SD (*n* = 6). *, *p* < 0.05; **, *p* < 0.01 compared with the vehicle group.

**Figure 6 biomolecules-10-00071-f006:**
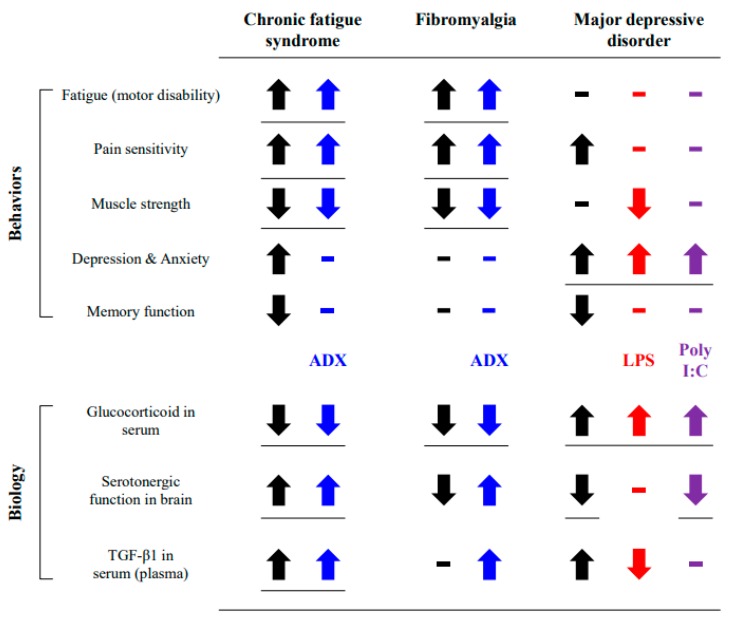
Summary of results. The obtained results were compared to the behavioral and biological features of CFS or similar diseases.

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
