# Peer review of "An Adrenalectomy Mouse Model Reflecting Clinical Features for Chronic Fatigue Syndrome"

_biomolecules, 2020, doi:10.3390/biom10010071_

Round 1
Reviewer 1 Report
Manuscript ID: biomolecules-678086
Title: An adrenalectomy mouse model reflecting clinical features for chronic fatigue syndrome
The paper by Jin-Seok Lee, Yoo-Jin Jeon, Samuel-Young Park and Chang-Gue Son tries to show the features of the LPS E.coli O111:B4., poly 287 I:C, and ADX mouse models in comparison with the major pathophysiological features of CFS.
This article can be an important consideration for many investigators, but there are some comments that need to be addressed by authors :
1) The introduction presents the literature background with some gaps in the literature.
2) Similarly to the introduction, there is a gap in the literature in the section of the discussion. The discussion presents some past data but lacks in-depth analysis and discussion of the study data.
3) Please mark in important parts of the text that your LPS is from E.coli serotype O111:B4. Please write something about LPS from different species and serotypes of bacteria in the paragraphs of your manuscript. Neuroimmunological disturbances, under the influence of LPS, depending on its origin (not only within the species, but also the serotype of bacteria), may contribute to pathological processes associated with various diseases, starting with Alzheimer’s or Parkinson’s disease, and ending with schizophrenia and depression.
Even the use of a low dose of LPS in the in vitro studies involving LPS derived from three different Salmonella serotypes did not change the number of DRG neurons but affected their neurochemistry, an example being the activity of LPS from S. Enteritidis resulting in an increase in the percentage of SP-positive neurons, while LPS from S.Minnesota and LPS from S.Typhimurium exerted the contrary effect [Mikołajczyk et al., IJMS. 2018, 19(9) DOI: 10.3390/ijms19092551]. So It should be noted that the activity of even low doses of LPS on the neurons may vary and may depend not only on the species but also on the serotype of the bacteria.
4) Please discuss the problem of LPS time duration in organisms (in the paragraphs of your manuscript). You can expect the different activity of LPS for example one day after LPS injection and three or seven days after LPS administration, especially in the nervous system. Moreover, the presence of LPS S. Enteritidis in the body even in the amount which does not induce any symptoms of illness may bring about unknown long-term consequences associated with its action on the cells and biologically active substances in the nervous, immune and endocrine systems. [Mikołajczyk A., Złotkowska D. (2018). IJMS19(10), DOI: 10.3390/ijms19103274; Mikołajczyk A., Złotkowska D. 2019 Toxins, 2019, 11(2), DOI:10.3390/toxins11020091]. So also please discuss the problem of LPS dose.
5) Please write in the section of materials and methods about post-operative (analgesic care and other drugs) treatment.
6) The details regarding pre-operative sedation, main anesthesia and post-operative treatment are a must.
7) "These mice were 86 then allowed to recover for 7 days along and were supplied with 1% glucose in tap water to maintain osmotic pressure." Please give some position of literature to affirm that mice recover 7 days after adrenalectomy.
8) Please write about post-operative (analgesic care and other drugs) treatment and their influence on the organism (some drugs can influence n changes some even 14 days /or longer/ after their administration).
Please approach problems from a broader perspective, involving variety of different LPS activity and time duration as well. The potential readers of this paper should see an explanation of choice of LPS E.coli and information that hypothetically LPS from other serotypes (or different doses or time duration) maybe can be useful as CFS models.
Author Response
Reviewer 1
Comments and Suggestions for Authors
Manuscript ID: biomolecules-678086
Title: An adrenalectomy mouse model reflecting clinical features for chronic fatigue syndrome
The paper by Jin-Seok Lee, Yoo-Jin Jeon, Samuel-Young Park and Chang-Gue Son tries to show the features of the LPS E. coli O111:B4., poly I:C, and ADX mouse models in comparison with the major pathophysiological features of CFS.
This article can be an important consideration for many investigators, but there are some comments that need to be addressed by authors:
1) The introduction presents the literature background with some gaps in the literature.
=> We sincerely appreciate for reviewer’ helpful and insightful comment. According to reviewer suggestion, we re-described some sentences with additional recent literatures to minimize gaps in ‘introduction section’ of revised manuscript.
2) Similarly to the introduction, there is a gap in the literature in the section of the discussion. The discussion presents some past data but lacks in-depth analysis and discussion of the study data.
=> As reviewer suggested, lacks of analysis and interpretation in ‘discussion section’ were supplemented by additional description and references in the current version of revised manuscript.
3) Please mark in important parts of the text that your LPS is from E. coli serotype O111:B4. Please write something about LPS from different species and serotypes of bacteria in the paragraphs of your manuscript. Neuroimmunological disturbances, under the influence of LPS, depending on its origin (not only within the species, but also the serotype of bacteria), may contribute to pathological processes associated with various diseases, starting with Alzheimer’s or Parkinson’s disease, and ending with schizophrenia and depression.
Even the use of a low dose of LPS in the in vitro studies involving LPS derived from three different Salmonella serotypes did not change the number of DRG neurons but affected their neurochemistry, an example being the activity of LPS from S. Enteritidis resulting in an increase in the percentage of SP-positive neurons, while LPS from S.Minnesota and LPS from S.Typhimurium exerted the contrary effect [Mikołajczyk et al., IJMS. 2018, 19(9) DOI: 10.3390/ijms19092551]. So It should be noted that the activity of even low doses of LPS on the neurons may vary and may depend not only on the species but also on the serotype of the bacteria.
=> We really appreciate for reviewer’ professional comment. As reviewer mentioned, the serotype and origin must be distinct. We additionally described the origin of LPS (from E. coli) and importance of LPS serotype with references (doi: 10.3390/ijms19092551 and 10.1016/s0024-3205(00)00821-3) in revised manuscript.
4) Please discuss the problem of LPS time duration in organisms (in the paragraphs of your manuscript). You can expect the different activity of LPS for example one day after LPS injection and three or seven days after LPS administration, especially in the nervous system. Moreover, the presence of LPS S. Enteritidis in the body even in the amount which does not induce any symptoms of illness may bring about unknown long-term consequences associated with its action on the cells and biologically active substances in the nervous, immune and endocrine systems. [Mikołajczyk A., Złotkowska D. (2018). IJMS19(10), DOI: 10.3390/ijms19103274; Mikołajczyk A., Złotkowska D. 2019 Toxins, 2019, 11(2), DOI:10.3390/toxins11020091]. So also please discuss the problem of LPS dose.
=> According to reviewer suggestion, we detailedly discussed the importance of serotype, dose, and duration after LPS injection with references (doi: 10.3390/ijms19103274, 10.4161/viru.26083, and 10.1371/journal.pone.0201375) in revised manuscript.
5) Please write in the section of materials and methods about post-operative (analgesic care and other drugs) treatment.
=> Thank reviewer for the comment. We added the detailed description for adrenalectomy and post-operative procedure in the ‘material and methods section’ of revised manuscript.
6) The details regarding pre-operative sedation, main anesthesia and post-operative treatment are a must.
=> We thoroughly understand reviewer’s concern. Neuro-inflammatory features are key factors in this study; however general analgesic drugs can influence those makers. For example, NSAIDs have potential anti-inflammatory effects systemically (doi: 10.3389/fnagi.2010.00035 and 10.14336/AD.2017.0306), and meanwhile opioids exacerbate microglia-derived neuroinflammation (doi: 10.1016/j.cobeha.2016.12.003). So, we adapted the analgesic effects by ketamine only during surgery (J Am Assoc Lab Anim Sci. 2016 Nov;55(6):794-800) and some post-operative care.
7) "These mice were then allowed to recover for 7 days along and were supplied with 1% glucose in tap water to maintain osmotic pressure." Please give some position of literature to affirm that mice recover 7 days after adrenalectomy.
=> We fully agree with reviewer opinion. A lot of previous studies indicated that mice were allowed to recovery for at least 10 days. Nonetheless, some studies choose the recovery period for 7 days after adrenalectomy (doi: 10.1155/2013/809062, 10.1681/ASN.2017030243, and 10.1016/j.jid.2017.02.984). We decided recovery period based on supporting preliminary data includes body weight and food intake. Besides, the population of blood cells were not different from sham-operated mice, as indicated in Suppl. Table 1.
8) Please write about post-operative (analgesic care and other drugs) treatment and their influence on the organism (some drugs can influence n changes some even 14 days /or longer/ after their administration).
=> As reviewer mentioned, some analgesic agents or post-operative care could contribute to unexpected changes. As we explains above, neuro-inflammatory features are key factors in this study, And then, general analgesic drugs such as NSAIDs or opioids are known to affect neuro-inflammatory conditions. So, we tried to minimize those effects via adaption of the analgesic effects by ketamine only during surgery and some post-operative care. We additionally discussed about influence of surgical conditions on the behavioral results.
Please approach problems from a broader perspective, involving variety of different LPS activity and time duration as well. The potential readers of this paper should see an explanation of choice of LPS E.coli and information that hypothetically LPS from other serotypes (or different doses or time duration) maybe can be useful as CFS models.
=> As discussed above, we further explained the rationale of LPS condition including origin, serotype, dose, and duration in ‘discussion section’ of revised manuscript.
Reviewer 2 Report
The manuscript entitled “An adrenalectomy mouse model reflecting clinical features for chronic fatigue syndrome” is a well written and discussed topic. The manuscript could be useful for the readers of “Biomolecules” journal. Though, I have some minor comments.
Introduction 2nd paragraph, please explain the importance of CFS animal model? Section 2.2, the sentence “These mice were then allowed to recover for 7 days along and were supplied with 1% glucose in tap water to maintain osmotic pressure.” looks confusing. to whom osmotic pressure maintained and why? Figure 1 legend, to make it more precise, please replace "animal" with mice models. Figure 2 please increase the font size of the scale bar of fluorescent images and figure bar legends of figure 2B. Please work with figures resolution as they look blurred. Explain "IOM" in line discussion line 305.
Author Response
Reviewer 2
Comments and Suggestions for Authors
The manuscript entitled “An adrenalectomy mouse model reflecting clinical features for chronic fatigue syndrome” is a well written and discussed topic. The manuscript could be useful for the readers of “Biomolecules” journal. Though, I have some minor comments.
Introduction 2nd paragraph, please explain the importance of CFS animal model?
=> We sincerely appreciate for reviewer’ professional comment. We additionally described the importance of CFS animal model in the ‘introduction section’ of revised manuscript.
Section 2.2, the sentence “These mice were then allowed to recover for 7 days along and were supplied with 1% glucose in tap water to maintain osmotic pressure.” looks confusing. to whom osmotic pressure maintained and why?
=> We really apology for the our misdescriptions. Actually, there is some misunderstand with one participating author (contribute to Methodology). To compensate for the loss of aldosterone (osmolarity), we used the 0.9% saline and 1% glucose in tap water. It was revised in current version of manuscript.
Figure 1 legend, to make it more precise, please replace "animal" with mice models.
=> Thank reviewer for the help. We revised it.
Figure 2 please increase the font size of the scale bar of fluorescent images and figure bar legends of figure 2B. Please work with figures resolution as they look blurred.
=> According to reviewer suggestions, we revised all imaging issues.
Explain "IOM" in line discussion line 305.
=> The full name of ‘IOM’ is addressed in discussion line 297.
Round 2
Reviewer 1 Report
I recommend the acceptance of this manuscript, but please change the phrases „lipopolysaccharide (LPS)” into „lipopolysaccharide (LPS) from Escherichia coli O111:B4" in the abstract.